# Revisiting Image Classifier Training for Improved Certified Robust Defense against Adversarial Patches

**Aniruddha Saha**[*]                                                    *anisaha1@umd.edu*
*University of Maryland, College Park*

**Shuhua Yu**[*]                                                         *shuhuay@andrew.cmu.edu*
*Carnegie Mellon University*

**Mohammad Sadegh Norouzzadeh**                          *arash.norouzzadeh@us.bosch.com*
*Bosch Center for AI*

**Wan-Yi Lin**                                                          *wan-yi.lin@us.bosch.com*
*Bosch Center for AI*

**Chaithanya Kumar Mummadi**                          *chaithanyakumar.mummadi@de.bosch.com*
*Bosch Center for AI*

**Reviewed on OpenReview:** *https://openreview.net/forum?id=2tdhQMLg36*

## Abstract

Certifiably robust defenses against adversarial patches for image classifiers ensure correct prediction against any changes to a constrained neighborhood of pixels. PatchCleanser Xiang et al. (2022), the state-of-the-art certified defense, uses a double-masking strategy for robust classification. The success of this strategy relies heavily on the model's invariance to image pixel masking. In this paper, we take a closer look at model training schemes to improve this invariance. Instead of using Random Cutout DeVries & Taylor (2017) augmentations like PatchCleanser, we introduce the notion of worst-case masking, i.e., selecting masked images which maximize classification loss. However, finding worst-case masks requires an exhaustive search, which might be prohibitively expensive to do on-the-fly during training. To solve this problem, we propose a two-round greedy masking strategy (Greedy Cutout) which finds an approximate worst-case mask location with much less compute. We show that the models trained with our Greedy Cutout improves certified robust accuracy over Random Cutout in PatchCleanser across a range of datasets and architectures. Certified robust accuracy on ImageNet with a ViT-B16-224 model increases from 58.1% to 62.3% against a 3% square patch applied anywhere on the image.

## 1 Introduction

Deep learning based image classifiers are vulnerable to adversarial patches applied to the input Brown et al. (2017); Eykholt et al. (2018), where image pixels bounded within a confined connected region, usually square or circular, can be adversarially crafted to induce misclassification. Patch attacks are formulated in this way to mimic printing and placing an adversarial object in the scene. It is easier to realize a patch attack in the physical world compared to full image adversarial perturbation, as the latter might need a compromise in the compute system. Therefore, adversarial patch attacks pose a significant threat to real-world computer vision systems.

---

[*]Equal contribution. Work done during internship at Bosch Center for AI, Pittsburgh.

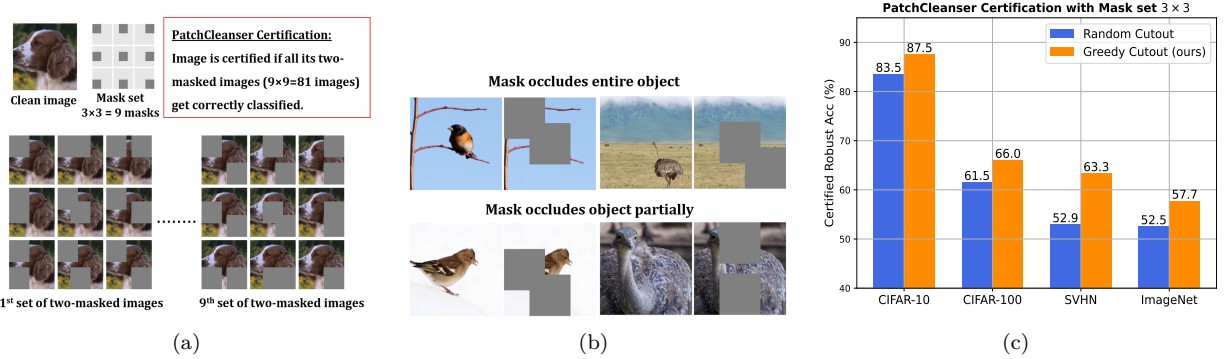

Figure 1: **Better invariance to pixel masking improves certified robustness to adversarial patches** (a) An illustration of PatchCleanser Xiang et al. (2022) double masking strategy. (b) Images are hard to classify when masks occlude objects of interest. Top row: Certification is not possible when mask completely occludes the object. Bottom row: Random Cutout failed to enable certification in cases of partial occlusion, whereas our *Greedy Cutout* succeeds. (c) *Greedy Cutout* improves PatchCleanser certification by a large margin. Certified robust accuracy on ImageNet with a ViT-B16-224 model increases from 52.5% to 57.7% against a 3% square patch ( $\approx 39 \times 39$) applied anywhere on the image with $\mathbf{M}_{3 \times 3}$.

Defenses against patch attacks fall into two categories: empirical and certified defense. Empirical defenses Rao et al. (2020); Hayes (2018); Mu & Wagner (2021) often utilize robust training which incorporates adversarial inputs generated from specific attacks to improve robustness. But they are susceptible to attacks unseen during training. On the other hand, certified defenses guarantee correct predictions against any adaptive white-box attacker under a given threat model. Hence, the certified robust accuracy is the guaranteed lower bound of the model performance. Prior works Levine & Feizi (2020); Lin et al. (2021); Metzen & Yatsura (2021); Xiang et al. (2022) acknowledge this strong robustness property of certified defense and propose different certification procedures to improve the certified robustness. Note that, we consider a single adversarial square patch applied on an image in our threat model in this work.

PatchCleanser Xiang et al. (2022) is the state-of-the-art certified defense method against patch attacks. Similar to other certified methods, it assumes the defense has a conservative estimate of the patch size (e.g. 3% pixels) and thereby designs a mask set - masks with certain size placed at reserved spatial locations on an image. Fig 1(a) shows a mask set $\mathbf{M}_{3 \times 3}$ of 9 masks, each with size $100 \times 100$, that are applied on $224 \times 224$ image. This mask set is designed to ensure that an adversarial patch of an estimated size (e.g. 3% pixels - $39 \times 39$ patch) will be covered and therefore neutralized by at least one of the masks. The method proposes two-round pixel masking defense and certifies an image if predictions of all its double-masked images agree with the ground truth. An image is certified if all the two-masked $9^2 = 81$ combinations get correctly classified. More details about this method can be found in Sec 3.3.

PatchCleanser shows that a classifier trained on two-masked (or double-masked) images using random Cutout DeVries & Taylor (2017) augmentation improves certification accuracy. At training time, a two-masked image is obtained by applying random Cutout twice on a clean image (see Fig 4). During certification, we observe that some of the two-masked images are hard to classify as the masks completely occlude the object of interest beyond recognition as shown in Fig 1(b) (top row). It is not possible to certify such images with the considered mask set. Besides, we made a visual observation that random Cutout trained classifiers even misclassify some two-masked images with preserved semantic clues, see bottom row of Fig 1(b). Misclassification on such images result in underestimating classifier's certified robustness. We consider that random Cutout augmentation is not fully effective or invariant towards pixel masking and limits classifier potential to leverage the defense. This observation motivates our work to explore the training schemes that improve invariance to pixel masking and thereby enhance classifer's certified robustness through PatchCleanser.

We revisit image classifier training strategy to particularly target the classifier's failure modes during Patch-Cleanser certification. The masks of misclassified images during certification would serve as possible worst-

case masks i.e. masks that induce image misclassification. Instead of Random Cutout augmentation, we propose to choose worst-case masked images with high classification loss for training. The worst-case masks are chosen from the mask set that are used during certification. Finding such worst-case masks from mask set $\mathbf{M}_{3 \times 3}$ and $\mathbf{M}_{6 \times 6}$ exhaustively requires $9 \times 9 = 81$ and $36 \times 36 = 1296$ two-mask forward passes respectively, which is prohibitively expensive to do on-the-fly during training. To this end, we propose a simple but effective greedy masking strategy called *Greedy Cutout* to approximate the worst-case masks with 42% and 96% less computation overhead than the grid search on $\mathbf{M}_{3 \times 3}$ and $\mathbf{M}_{6 \times 6}$ respectively. We propose to train the image classifiers on the *Greedy Cutout* images and demonstrate that such masking strategy improves certified robustness across different datasets as shown in Fig 1(c).

**We summarize our contributions as follows:** We observe that training the classifier with standard Cutout augmentation is not entirely effective for all two-masked images. Therefore, we take a closer look at the invariance of image classifiers to pixel masking with an objective to improve PatchCleanser certification. We propose a simple *Greedy Cutout* strategy to train the classifier with worst-case masked images. We compare with various other masking strategies like Random Cutout DeVries & Taylor (2017), Cutout guided by certification masks, Gutout (Cutout guided by Grad-CAM) Choi et al. (2020), Cutout with exhaustive search (refer Sec. 4) and show that our training strategy improves certified robustness over other baselines across different datasets and architectures.

## 2    Related Work

**Certified defenses against patch attack:** Certified defenses against adversarial patch for image classifiers provide guaranteed robust accuracy lower bound. Because patch attack has maximum strength within the patch region, certified defenses either ensembles from local regions Levine & Feizi (2020); Salman et al. (2022); Lin et al. (2021) or use models with small receptive field and masking Metzen & Yatsura (2021); Xiang et al. (2021) to reduce global influence of the patches. Current state-of-the-art PatchCleanser Xiang et al. (2022) combines masking and ensembling by aggregating predictions of specifically-designed masked copies and shows significant improvement over previous methods. Our work follows the certification and inference procedure of PatchCleanser Xiang et al. (2022). However, we show that worst-case mask augmentation can improve invariance to pixel masking and consequently the certified robust accuracy of PatchCleanser.

**Augmentation for adversarial robustness:** Data augmentations help in training smooth and generalizable models, and are used to train adversarially-robust classifiers. Adversarial training Madry et al. (2018), Niu et al. (2022) which augments input with gradient given a model has been established as one of the most powerful approach for training robust models. Studies have also shown that adversarial training improves generalizability of models Yi et al. (2021); Mao et al. (2022). Gradient-free augmentation has shown improvements in model robustness: Gowal et al. (2021) utilizes unconditional generative models to generate an additional training dataset, and Li & Spratling (2023) proposed padding before cropping (PadCrop) for augmentation to reduce robust overfitting. Cutout DeVries & Taylor (2017), randomly masks out square regions of input images during training to improve clean accuracy. We propose a variation to random Cutout augmentation scheme in our paper to improve certified robustness.

## 3    Method

### 3.1    Patch attack formulation

This paper focuses on adversarially-robust image classification. We use $\mathcal{X} \subset [0,1]^{W \times H \times C}$ to denote an image, where each image has width $W$, height $H$, number of channels $C$, and pixel values in $[0,1]$. We use $\mathcal{Y}$ to denote the label set, and $\mathbb{F} : \mathcal{X} \to \mathcal{Y}$ denotes the image classification model. Note that we do not make any assumptions about the network architecture, and thus we present model-agnostic method.

We consider patch attack, denoted by $\mathcal{A}$, that can arbitrarily manipulate the pixels within a square region covering up to 3% of image area. We use $\mathbf{r} \in \{0,1\}^{W \times H}$ to denote a binary mask with the aforementioned square region, where pixels within the square region is set to 0 while others are set to 1. Since the region can be placed in any location of an image, we denote all such square regions covering less than 3% of an image

as $\mathcal{R}$. Formally, for an image $\mathbf{x}$ with true label $y$, the patch attack $\mathcal{A}$ can generate a set of images under attack, i.e., $\mathcal{A}_{\mathcal{R}}(\mathbf{x}) = \{\mathbf{r} \odot \mathbf{x} + (1 - \mathbf{r}) \odot \mathbf{x}' \mid \mathbf{x}' \in \mathcal{X}, r \in \mathcal{R}\}$, where the attack aims to find an image $\hat{\mathbf{x}} \in \mathcal{A}_{\mathcal{R}}(\mathbf{x})$ such that $\mathbb{F}(\hat{\mathbf{x}}) \neq y$. Here, $\odot$ represents pixel-wise multiplication.

## 3.2 Defense objectives

The goal of certifiable defense is to build a robust model $\mathbb{F}$ such that for a given image label pair $(\mathbf{x}, y)$, we have $\forall \hat{\mathbf{x}} \in \mathcal{A}_{\mathcal{R}}(\mathbf{x}), \mathbb{F}(\hat{\mathbf{x}}) = y$, then $\mathbf{x}$ is termed as a *certified image* and the fraction of certified images in the test data is defined as *certified robust accuracy*. We consider the process to determine whether an image-label pair $(\mathbf{x}, y)$ can be certified follows PatchCleanser Xiang et al. (2022). Certified robust accuracy provides a lower bound on model's classification accuracy under patch attack. On the other hand, we want to avoid deteriorated performance over clean data free of patch attacks. Therefore, the other metric we wish to maintain is the fraction of image label pair $(\mathbf{x}, y)$ satisfying $\mathbb{F}(\mathbf{x}) = y$ over test data, which we refer it as *clean accuracy*. Computing clean accuracy requires inference process sweeping over the test data.

## 3.3 Background on PatchCleanser

In this work, we follow the inference and certification procedures of PatchCleanser Xiang et al. (2022). PatchCleanser achieves certified defense via double-masking strategy. A mask $\mathbf{m} \in \{0, 1\}^{W \times H}$ is defined as a binary image with the same dimension as images in $\mathcal{X}$ where pixels within the mask take value 0, and the rest take value 1, and thus a masked image $\mathbf{x} \odot \mathbf{m}$ will exhibit an artificially occluded region within the mask while other pixels remains unaffected. PatchCleanser crafts a well-designed set of masks $\mathbf{M}$ and provides a provable certification procure based on predicted classes over the set of masked images $\{\mathbf{x} \odot \mathbf{m} \mid \mathbf{m} \in \mathbf{M}\}$.

PatchCleanser designs the mask set $\mathbf{M}$ satisfying $\mathcal{R}$-*covering* property, that is, for any adversarial patch, there exists at least one mask $\mathbf{m} \in \mathbf{M}$ that would completely cover the adversarial patch. Formally, $\forall \mathbf{r} \in \mathcal{R}, \exists \mathbf{m} \in \mathbf{M}$, s.t. $\mathbf{m}[i, j] \leq \mathbf{r}[i, j], \forall i, j$. PatchCleanser proves that if an image label pair $(\mathbf{x}, y) \in \mathcal{X} \times \mathcal{Y}$ satisfies that $\forall \mathbf{m}_1, \mathbf{m}_2 \in \mathbf{M} \times \mathbf{M}, \mathbb{F}(\mathbf{x} \odot \mathbf{m}_1 \odot \mathbf{m}_2) = y$, then $(\mathbf{x}, y)$ is certified. Therefore, improving the certified robust accuracy reduces to maintain high classification accuracy for $\mathbb{F}$ on masked images $\mathbf{x} \odot \mathbf{m}_1 \odot \mathbf{m}_2, \mathbf{m}_1, \mathbf{m}_2 \in \mathbf{M} \times \mathbf{M}$. To this end, PatchCleanser employs Cutout data augmentation DeVries & Taylor (2017) to finetune model $\mathbb{F}$.

For inference, PatchCleanser first obtains all one-mask predictions $\mathbb{F}(\mathbf{x} \odot \mathbf{m}), \forall \mathbf{m} \in \mathbf{M}$. If the image $\mathbf{x}$ has no adversarial patch, then all one-mask predictions should be the same as $\mathbb{F}(\mathbf{x})$, in this case it outputs the agreed prediction as the predicted class for $\mathbf{x}$. If $\mathbf{x}$ contains an adversarial patch that induces misclassification, and one of the masks $\mathbf{m} \in \mathbf{M}$ completely cover or occlude the adversarial patch, then at least one prediction should be different from the rest. For each disagreed (minority) single-masked image prediction, the masked image is proceeded to second-round masking with the same mask set $\mathbf{M}$ and all the two-masked images are classified with $\mathbb{F}$. If the single-masked image has fully occluded the adversarial patch, then all its two-masked images also fully occlude the adversarial patch, yielding unanimous predictions from the two-masked images as the final output of $\mathbb{F}(\mathbf{x})$. If none or more than one disagreed case have unanimous two-masked predictions, then none of the classes should be trusted, and the majority predicted class of the single-masked images is the final prediction of $\mathbf{x}$. Example masked copies can be found in Figure 1(a). PatchCleanser shows that if an image-label pair $(\mathbf{x}, y)$ is certified, the above inference procedure will always output the correct label $y$.

## 3.4 Description of our method - Greedy Cutout

Recall that the classifier $\mathbb{F}$ should be fine-tuned to double-masked images $\{\mathbf{x} \odot \mathbf{m}_1 \odot \mathbf{m}_2 \mid \mathbf{m}_1, \mathbf{m}_2 \in \mathbf{M} \times \mathbf{M}\}$ so that it is less sensitive to presence of $\mathbf{m} \in \mathbf{M}$. For this purpose, we resort to augmented training data with $(\mathbf{x}', y)$ where $\mathbf{x}'$ is obtain from applying masks or combination of masks on $\mathbf{x}$. PatchCleanser uses Random Cutout augmentation DeVries & Taylor (2017), i.e., applying two masks of size $128 \times 128$ at random locations to $224 \times 224$ training images (see Fig. 4). In this work, we develop a systematic approach to find $\mathbf{x}'$ that are more empirically effective. The main strategy is to use several inference rounds to identify *worst-case* masks that lead to highest losses, then we apply these *worst-case* masks to $\mathbf{x}$ to obtain augmented data point $(\mathbf{x}', y)$ for training the model.

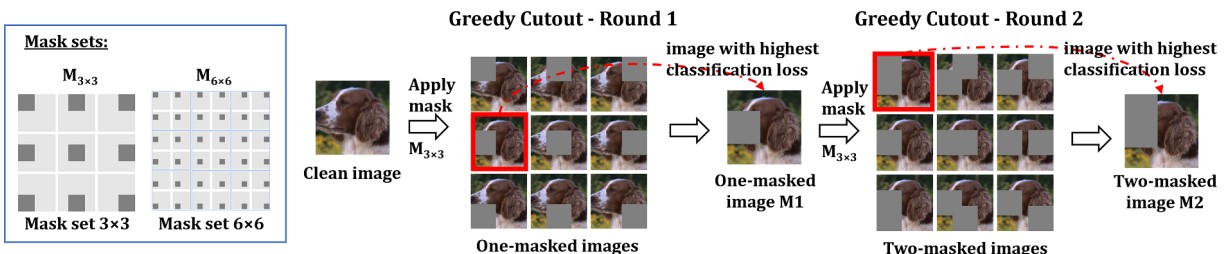

Figure 2: **Illustration of two-round *Greedy Cutout* masking -** Each round applies masks from the same mask set $\mathbf{M}_{k \times k}$ ($k = 3$ or $6$), making the number of forward passes for both rounds to $(9 \times 2) - 1 = 17$ and $(36 \times 2) - 1 = 71$ when $k = 3$ and $6$ respectively, to find the masked image with highest classification loss. The two-masked image $\mathbf{M}_2$ obtained from Round 2 is used for training the classifier.

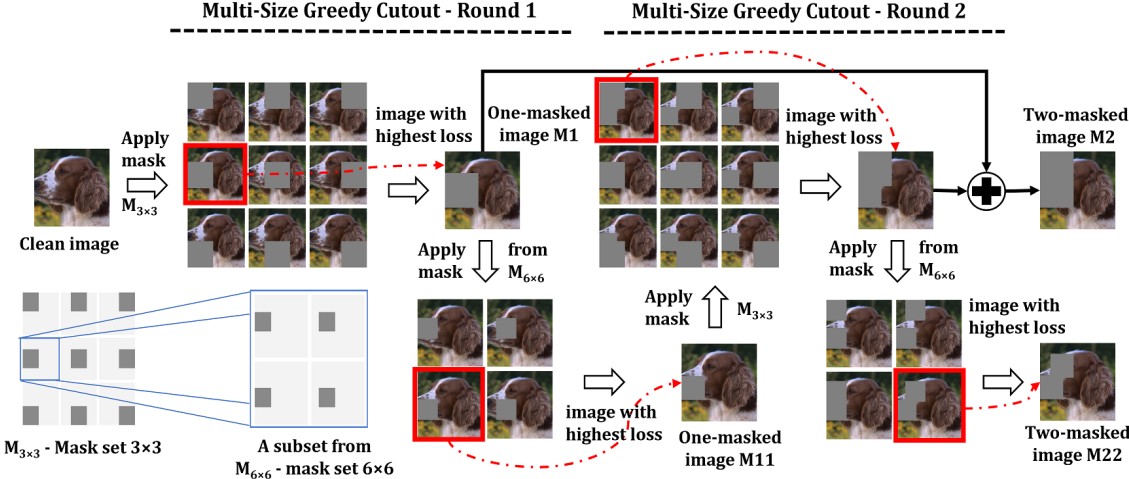

Figure 3: **Illustration of our multi-size two-round *Greedy Cutout* masking -** Each round begins with applying larger masks from $\mathbf{M}_{3 \times 3}$ with 9 forward passes, which narrow downs the search area for the smaller masks from $\mathbf{M}_{6 \times 6}$ to just 4 forward passes. With a total of 25 unique forward passes, the approximated worst-case masked images $\mathbf{M}_2$ and $\mathbf{M}_{22}$ from $\mathbf{M}_{3 \times 3}$ and $\mathbf{M}_{6 \times 6}$ are jointly used for training.

**Image and mask preparations:** For datasets of high-resolution images such as *ImageNet* and *ImageNette*, we resize and crop them to $224 \times 224$ before feeding them into model $\mathbb{F}$. For datasets with images with low-resolution such as CIFAR-10, we resize them to $224 \times 224$ via bicubic interpolation for a better classification performance. In this work, we generate sets of masks following PatchCleanserXiang et al. (2022). Concretely, we use two mask sets $\mathbf{M}_{3 \times 3}$ and $\mathbf{M}_{6 \times 6}$ (see the leftmost part of Figure 2 for an illustration, where light gray blocks represent image area and deep gray smaller blocks represent masks) that both satisfy $\mathcal{R}$-*covering* property: $\mathbf{M}_{3 \times 3}$ is a set of $3 \times 3$ masks with 9 masks in total, and each mask is of size $100 \times 100$; $\mathbf{M}_{6 \times 6}$ is a set of $6 \times 6$ masks with 36 masks, and every mask is in shape of $69 \times 69$. Note that, due to the careful choice of $\mathbf{M}_{3 \times 3}$ and $\mathbf{M}_{6 \times 6}$, each mask in $\mathbf{M}_{3 \times 3}$ can be fully covered by 4 masks in $\mathbf{M}_{6 \times 6}$. Empirically, $\mathbf{M}_{6 \times 6}$ leads to higher certified robust accuracy since two smaller masks have a smaller effect on the prediction on $\mathbb{F}$ than the larger ones. However, $\mathbf{M}_{6 \times 6}$ has $\approx 8$ times larger size than $\mathbf{M}_{3 \times 3}$ that requires more computation in both inference and certification. Striking a delicate trade-off between computational cost and accuracy is the core of our method.

**Greedy Cutout:** We present a baseline version of *Greedy Cutout* (See Figure 2 for an illustration). We first define the prediction loss of $\mathbb{F}$ on image label pair $(\mathbf{x}, y)$. Suppose within the model of $\mathbb{F}$, for each label $y' \in \mathcal{Y}$, the confidence level for $\mathbb{F}$ to predict $y'$ is $p_{y'}$, then we use the cross entropy loss $\ell(\mathbb{F}(\mathbf{x}, y)) = - \sum_{y \in \mathcal{Y}} c_{y'} \log p_{y'}$,

where $c_{y'} = 1$ for $y' = y$ otherwise $c_{y'} = 0$. For mask set $\mathbf{M}_{k \times k}(k = 3$ or $k = 6)$, we could identify the worst-case mask combination $\mathbf{m}_1, \mathbf{m}_2 \in \mathbf{M}_{k \times k} \times \mathbf{M}_{k \times k}$ that incurs the largest loss $\ell(\mathbb{F}(\mathbf{x} \odot \mathbf{m}_1 \odot \mathbf{m}_2, y)$ with grid search, but inferencing over all $\binom{k^2}{2} + k^2$ (45 when $k = 3$, and 666 when $k = 6$) unique combinations would be computationally prohibitive, thus motivating the *Greedy Cutout* strategy.

By Greedy Cutout, we find the worst-case mask in each individual masking round. In the first round, for each $\mathbf{m}_1 \in \mathbf{M}_{k \times k}$, we compute the loss $\ell(\mathbb{F}(\mathbf{x} \odot \mathbf{m}_1), y)$, and denote the mask incurring the highest loss as $\mathbf{m}_1^*$. Then, in the second round, for each $\mathbf{m}_2 \in \mathbf{M}_{k \times k}, \mathbf{m}_2 \neq \mathbf{m}_1$, we compute the loss $\ell(\mathbb{F}(\mathbf{x} \odot \mathbf{m}_1^* \odot \mathbf{m}_2), y)$, and find the $\mathbf{m}_2^*$ with the highest loss. Then, we empirically use $\mathbf{M}_2 := \mathbf{m}_1^* \odot \mathbf{m}_2^*$ as the worst-case mask combination. Although $\mathbf{M}_2$ may not be the exact worst-case mask, $(\mathbf{x} \odot \mathbf{M}_2, y)$ still provides a guidance for fine-tuning. More importantly, with *Greedy Cutout*, we significantly reduce the inference burden down to $2k^2 - 1$ (17 when $k = 3$; and 71 when $k = 6$) compared to grid search. However, for $k = 6$, this computation cost is still high but this mask set has better accuracy over $\mathbf{M}_{3 \times 3}$. We then further develop *Multi-size Greedy Cutout* (See Fig 3) that benefits from the high accuracy of $\mathbf{M}_{6 \times 6}$ while keeping computation cost comparable with Greedy Cutout with $\mathbf{M}_{3 \times 3}$.

**Multi-size Greedy Cutout:** Similar to the above method, this variant also involves two rounds.

Round 1 - Given clean image $\mathbf{x}$, we first apply each mask in $\mathbf{M}_{3 \times 3}$ and inference on each image in $\{\mathbf{x} \odot \mathbf{m} \mid \mathbf{m} \in \mathbf{M}_{3 \times 3}\}$. We then find the mask that incurs highest loss $\ell(\mathbb{F}(\mathbf{x} \odot \mathbf{m}), y)$, and denote this mask as $\mathbf{M}_1$. Then, we find the collection of 4 masks in $\mathbf{M}_{6 \times 6}$ that fully covers $\mathbf{M}_1$. We apply the previous 4 masks from $\mathbf{M}_{6 \times 6}$ on clean image $\mathbf{x}$ and inference each masked image, then we store the mask incurs highest loss among the four, denote this mask as $\mathbf{M}_{11}$.

Round 2 - We apply each $\mathbf{m} \in \mathbf{M}_{3 \times 3}$ on $\mathbf{x} \odot \mathbf{M}_{11}$, and find the $\mathbf{m}_3^*$ that incurs largest highest loss among the 9 masks. Then we output our first mask $\mathbf{M}_2 = \mathbf{M}_1 \odot \mathbf{m}_3^*$. For $\mathbf{m}_3^*$, we identify the 4 masks from $\mathbf{M}_{6 \times 6}$ that fully covers $\mathbf{m}_3^*$, and apply the 4 masks on $\mathbf{x} \odot \mathbf{M}_{11}$ to find $\mathbf{m}_6^*$ that incurs highest loss. We output the second mask $\mathbf{M}_{22} = \mathbf{M}_{11} \odot \mathbf{m}_6^*$.

From the previous two rounds, we identify two *worst-case* masks $\mathbf{M}_2$ and $\mathbf{M}_{22}$. We only use in total $9+4 = 13$ inferences in each round so only 26 inferences in total. In our experiments, we show that this strategy uses similar amount of computation with baseline *Greedy Cutout* for $k = 3$, but achieves comparable performance of baseline *Greedy Cutout* for $k = 6$. We provide the pseudocode in the Algorithm 1 in the appendix, where two procedures ROUND-1 and ROUND-2 echo the *round 1* and *round 2* described above. Running Algorithm 1 with input $(\mathbf{x}, \mathbb{F}, \mathbf{M}_{3 \times 3}, \mathbf{M}_{6 \times 6})$ will lead to desired augmented data points $(\mathbf{x}_1', y)$ and $(\mathbf{x}_2', y)$ that can be used for fine-tuning the vanilla model $\mathbb{F}$.

# 4 Experiments

In this section, we describe our experimental setup, masking strategies and present the results. We use five popular image classification datasets: ImageNet Deng et al. (2009), ImageNette Howard et al. (2020), CIFAR-10 Krizhevsky et al. (2009), CIFAR-100 Krizhevsky et al. (2009), SVHN Netzer et al. (2011) and image classifiers from three different architecture families: ResNet He et al. (2016), ViT Dosovitskiy et al. (2020), ConvNeXt Liu et al. (2022). In particular, ResNetV2-50x1, ViT-B16-224 and ConvNeXt_tiny_in22ft1k pretrained models from timm library Wightman (2019). We finetune these pretrained models on different datasets using the proposed masking strategies for 10 epochs with SGD optimizer at learning rate 0.01 for ResNet and 0.001 for ViT, ConvNeXt and reduce the learning rate by a factor of 10 after 5 epochs. Images from all the datasets are resized to $224 \times 224$. We use batch size 128 for ImageNet and 64 for other datasets.

**Adversarial patches:** As in prior work Xiang et al. (2022); Chiang et al. (2019); Levine & Feizi (2020); Metzen & Yatsura (2021); Xiang et al. (2021), we report certified defense accuracy for square patches covering 1%, 2% and 3% of the input image for ImageNet, ImagNette and 0.4%, 2.4% for CIFAR-10, CIFAR-100, and SVHN. We allow patch perturbations to be unbounded (within image range) and consider a single square patch placed at any random location in the image.

Figure 4: **Illustration of different masking strategies -** We explore various strategies to improve the invariance of models towards pixel masking.

**Evaluation metrics:** We report two metrics: clean accuracy - amount of clean test images classified correctly by the PatchCleanser defense and certified robust accuracy - amount of test images pass through certification process i.e. a guaranteed lower bound on classifier robustness.

## 4.1 Improving invariance to pixel masking

The core idea of PatchCleanserXiang et al. (2022) is to mask image pixels to occlude and neutralize the adversarial patch. We visually observe that two masks that completely occlude the object make images harder to certify (see Fig. 1(b)). However, masked images with partial occlusions and preserved semantic cues stand a chance to get classified correctly. Hence, a crucial property that governs the success of certification is the classifier's *prediction invariance to pixel masking*. Below we explain different train time masking strategies that we investigate towards improving pixel masking invariance.

### 4.1.1 Random Cutout in PatchCleanser (PC-Cutout)

Random CutoutDeVries & Taylor (2017) to apply 2 masks of size 128×128 at random locations on a 224×224 image during training (see Fig. 4). Here, masks may be located partially beyond the image boundary. It is shown that the such Cutout augmentation improves certification over vanilla classifiers. We hypothesize that such Cutout strategy might not be fully effective to induce mask invariance and propose further improvement.

### 4.1.2 Random Cutout with certification masks

We investigate whether choosing random masks from certification mask sets, $\mathbf{M}_{3 \times 3}$ or $\mathbf{M}_{6 \times 6}$ (Section 3.4) improve certified robustness. $rand_{3 \times 3}$ strategy chooses the masks exclusively from $\mathbf{M}_{3 \times 3}$, whereas $rand_{6 \times 6}$ strategy chooses them from $\mathbf{M}_{6 \times 6}$. Moreover, we also employ the *rand* strategy where masks are randomly sampled from either of the sets $\mathbf{M}_{3 \times 3}$ or $\mathbf{M}_{6 \times 6}$ during training (see Fig. 4).

Table 1: **Clean and certified robust accuracy comparison -** We show the results of selected prior work in the top group of rows. Then we compare the performance of PatchCleanser(PC)-Cutout (Section 4.1.1), Random Cutout with certification masks (Section 4.1.2) and Multi-size (MS) Greedy Cutout (Section 4.1.5). We use three classifiers (ResNet, ViT, ConvNeXt), three datasets (ImageNette, ImageNet, CIFAR-10) and adversarial patch sizes in the range of 1% - 3% of image size. Underlined numbers represent highest robust accuracy obtained for a specific architecture, and bold numbers represent highest robust accuracy obtained across different architectures.

| Dataset | ImageNette | | | | | | ImageNet | | | | | | CIFAR-10 | | | |
|---|---|---|---|---|---|---|---|---|---|---|---|---|---|---|---|---|
| Patch size | 1% pixels | | 2% pixels | | 3% pixels | | 1% pixels | | 2% pixels | | 3% pixels | | 0.4% pixels | | 2.4% pixels | |
| Accuracy (%) | clean | robust | clean | robust | clean | robust | clean | robust | clean | robust | clean | robust | clean | robust | clean | robust |
| IBP Chiang et al. (2019) | - | - | - | - | - | - | - | - | - | - | - | - | 65.8 | 51.9 | 47.8 | 30.8 |
| CBN Zhang et al. (2020) | 94.9 | 74.6 | 94.9 | 60.9 | 94.9 | 45.9 | 49.5 | 13.4 | 49.5 | 7.1 | 49.5 | 3.1 | 84.2 | 44.2 | 84.2 | 9.3 |
| DS Levine & Feizi (2020) | 92.1 | 82.3 | 92.1 | 79.1 | 92.1 | 75.7 | 44.4 | 17.7 | 44.4 | 14.0 | 44.4 | 11.2 | 83.9 | 68.9 | 83.9 | 56.2 |
| PG-BN Xiang et al. (2021) | 95.2 | 89.0 | 95.0 | 86.7 | 94.8 | 83.0 | 55.1 | 32.3 | 54.6 | 26.0 | 54.1 | 19.7 | 84.5 | 63.8 | 83.9 | 47.3 |
| PG-DS Xiang et al. (2021) | 92.3 | 83.1 | 92.1 | 79.9 | 92.1 | 76.8 | 44.1 | 19.7 | 43.6 | 15.7 | 43.0 | 12.5 | 84.7 | 69.2 | 84.6 | 57.7 |
| BagCert Metzen & Yatsura (2021) | – | – | – | – | – | – | 45.3 | 27.8 | 45.3 | 22.7 | 45.3 | 18.0 | 86.0 | 72.9 | 86.0 | 60.0 |
| ViT-B Salman et al. (2022) | – | – | – | – | – | – | 73.2 | 43.0 | 73.2 | 38.2 | 73.2 | 34.1 | 90.8 | 78.1 | 90.8 | 67.6 |
| *Certification with mask set* $M_{3\times3}$ | | | | | | | | | | | | | | | | |
| **ResNet** | | | | | | | | | | | | | | | | |
| PC-Cutout Xiang et al. (2022) | 99.3 | 94.0 | 99.4 | 92.8 | 99.2 | 91.6 | 80.9 | 52.3 | 80.9 | 48.7 | 80.8 | 45.8 | 97.3 | 81.3 | 96.9 | 73.0 |
| Rand Cutout with cert. masks | 99.1 | 94.2 | 99.2 | 92.5 | 99.2 | 91.3 | 81.2 | 47.1 | 81.2 | 47.1 | 81.0 | 45.1 | 97.5 | 80.1 | 96.8 | 71.0 |
| MS Greedy Cutout (Ours) | 99.4 | 95.5 | 99.3 | 94.5 | 99.2 | 93.7 | 80.5 | 56.4 | 80.3 | 52.8 | 80.1 | 51.2 | 97.4 | 85.5 | 96.7 | 79.2 |
| **ViT** | | | | | | | | | | | | | | | | |
| PC-Cutout Xiang et al. (2022) | 99.4 | 96.2 | 99.4 | 95.1 | 99.4 | 94.0 | 82.8 | 58.5 | 82.5 | 55.2 | 82.4 | 52.5 | 98.5 | 89.0 | 98.0 | 83.5 |
| Rand Cutout with cert. masks | 99.5 | 96.5 | 99.5 | 95.4 | 99.5 | 94.7 | 82.8 | 57.2 | 82.7 | 54.8 | 82.7 | 52.7 | 98.5 | 89.0 | 98.0 | 83.1 |
| MS Greedy Cutout (Ours) | 99.4 | **96.9** | 99.4 | **96.0** | 99.4 | **95.5** | 81.2 | **59.5** | 81.2 | **58.7** | 81.1 | **57.7** | 98.5 | **91.5** | 98.2 | **87.5** |
| **ConvNeXt** | | | | | | | | | | | | | | | | |
| PC-Cutout Xiang et al. (2022) | 99.5 | 96.1 | 99.5 | 95.1 | 99.3 | 94.4 | 81.6 | 55.6 | 81.5 | 52.5 | 81.6 | 49.8 | 97.9 | 85.0 | 97.4 | 77.9 |
| Rand Cutout with cert. masks | 99.4 | 95.8 | 99.4 | 94.9 | 99.3 | 94.5 | 81.7 | 55.2 | 81.7 | 51.7 | 81.6 | 49.2 | 97.9 | 84.6 | 97.6 | 76.9 |
| MS Greedy Cutout (Ours) | 99.5 | 96.5 | 99.4 | 95.9 | 99.4 | 95.4 | 80.5 | 59.1 | 80.4 | 56.0 | 80.2 | 53.6 | 97.7 | 88.5 | 97.4 | 82.7 |
| *Certification with mask set* $M_{6\times6}$ | | | | | | | | | | | | | | | | |
| **ResNet** | | | | | | | | | | | | | | | | |
| PC-Cutout Xiang et al. (2022) | 99.7 | 96.6 | 99.6 | 95.2 | 99.4 | 93.9 | 81.3 | 59.2 | 81.2 | 54.3 | 81.1 | 50.9 | 97.7 | 86.1 | 97.5 | 77.8 |
| Rand Cutout with cert. masks | 99.7 | 96.5 | 99.5 | 95.0 | 99.4 | 93.8 | 81.6 | 58.6 | 81.6 | 53.1 | 81.5 | 50.3 | 97.9 | 86.8 | 97.6 | 77.1 |
| MS Greedy Cutout (Ours) | 99.6 | 97.8 | 99.5 | 96.4 | 99.4 | 95.6 | 81.0 | 63.9 | 80.9 | 58.9 | 80.8 | 56.6 | 97.7 | 90.8 | 97.5 | 84.3 |
| **ViT** | | | | | | | | | | | | | | | | |
| PC-Cutout Xiang et al. (2022) | 99.7 | 97.9 | 99.6 | 97.0 | 99.6 | 96.6 | 83.4 | 65.2 | 83.2 | 61.2 | 83.1 | 58.1 | 98.8 | 93.4 | 98.6 | 87.9 |
| Rand Cutout with cert. masks | 99.8 | 97.9 | 99.8 | 96.9 | 99.7 | 96.4 | 83.4 | 64.0 | 83.4 | 60.5 | 83.3 | 58.3 | 98.9 | 94.1 | 98.7 | 88.4 |
| MS Greedy Cutout (Ours) | 99.6 | **98.5** | 99.6 | **97.9** | 99.5 | **97.3** | 82.0 | **65.6** | 82.0 | **63.1** | 82.0 | **62.3** | 99.0 | **95.3** | 98.8 | **92.0** |
| **ConvNeXt** | | | | | | | | | | | | | | | | |
| PC-Cutout Xiang et al. (2022) | 99.8 | 98.3 | 99.8 | 97.3 | 99.7 | 96.5 | 82.1 | 63.1 | 82.2 | 58.3 | 82.1 | 55.0 | 98.2 | 90.7 | 98.0 | 82.8 |
| Rand Cutout with cert. masks | 99.7 | 98.2 | 99.8 | 96.9 | 99.6 | 96.4 | 82.3 | 62.7 | 82.3 | 57.1 | 82.2 | 54.5 | 98.3 | 90.3 | 98.2 | 81.9 |
| MS Greedy Cutout (Ours) | 99.7 | 98.7 | 99.7 | 97.9 | 99.6 | 97.4 | 81.3 | 66.5 | 81.1 | 61.8 | 81.0 | 59.4 | 98.1 | 93.3 | 97.8 | 87.3 |

### 4.1.3 Gutout - Cutout guided by Grad-CAM

Instead of Random Cutout, we examine the effectiveness of classifier training on saliency guided Cutout. We use image explanation heatmaps from literature to guide this process. Specifically, we use Grad-CAM Selvaraju et al. (2017) to find the most salient region of the image and selectively mask out the region using certification mask sets $M_{3\times3}$ or $M_{6\times6}$ in a two-round fashion (see Fig. 4 for illustration). We refer the masking strategies as $gutout_{3\times3}$, $gutout_{6\times6}$ and $gutout$ when training on the mask sets $M_{3\times3}$, $M_{6\times6}$, and both the mask sets. Such augmentation strategy was proposed in Choi et al. (2020) and we take inspiration from their implementation.

### 4.1.4 Cutout guided by exhaustive search

Cutout masks have edges which introduce sharp transitions from grayscale mask color to natural image color. These transitions can often happen over the object of interest and thus introduce artifacts which might make the learning problem harder. We believe that the two-masked images obtained from the saliency guided Cutout may not serve as the best training examples for the classifier to improve pixel masking invariance. To this end, we sweep through the entire search area by evaluating classifier performance on all possible two-masked image combinations from mask sets $M_{3\times3}$ or $M_{6\times6}$ and find the *worst-case masks* that result in highest classification loss. We train classifiers on the *worst-case masked images* and refer this masking strategy as $grid_{3\times3}$ and $grid_{6\times6}$ with with masks set $M_{3\times3}$ or $M_{6\times6}$ respectively. We believe that this

Table 2: **Results on additional datasets -** Clean and robust accuracy on CIFAR-100 and SVHN.

| Dataset | CIFAR-100 | | SVHN | |
|---|---|---|---|---|
| Patch size | 2.4% pixels | | 2.4% pixels | |
| Accuracy (%) | clean | robust | clean | robust |
| Certification with mask set $\mathbf{M}_{3\times3}$ | | | | |
| **ResNet** | | | | |
| PC-Cutout Xiang et al. (2022) | 86.8 | 45.9 | 93.8 | 43.4 |
| Multi size Greedy Cutout (Ours) | 85.8 | 53.9 | 93.4 | 54.5 |
| **ViT** | | | | |
| PC-Cutout Xiang et al. (2022) | 91.4 | 61.5 | 95.9 | 52.9 |
| Multi size Greedy Cutout (Ours) | 90.9 | **66.0** | 95.8 | **63.3** |
| Certification with mask set $\mathbf{M}_{6\times6}$ | | | | |
| **ResNet** | | | | |
| PC-Cutout Xiang et al. (2022) | 87.6 | 52.6 | 95.2 | 55.5 |
| Multi size Greedy Cutout (Ours) | 86.7 | 61.0 | 95.3 | 67.5 |
| **ViT** | | | | |
| PC-Cutout Xiang et al. (2022) | 92.4 | 69.1 | 97.2 | 67.5 |
| Multi size Greedy Cutout (Ours) | 92.1 | **74.5** | 97.1 | **76.8** |

approach is close to an upper bound on the best masking strategy. The downside is that it requires classifier evaluation on large number of all possible two-masked images, 45 unique two-masked images among $9^2 = 81$ in $\mathbf{M}_{3\times3}$ and 666 among $36^2 = 1296$ in $\mathbf{M}_{6\times6}$, thus makes this strategy computationally expensive.

### 4.1.5 Greedy Cutout

As mentioned above, the compute requirement of an exhaustive search for the *worst-case masked images* might be prohibitive to train the classifiers. On the other hand, light weight masking strategy like random Cutout or a heuristic like Gutout might not be fully effective for our objective. Our proposed masking strategy *Greedy Cutout* (Sec. 3.4) approximates the *worst-case masked images* with much less computation overhead, bringing the best of both worlds. We propose to train the classifiers with the *worst-case masked images* obtained from this masking strategy. We refer the masking strategy of our baseline version of *Greedy Cutout* as $greedy_{3\times3}$ and $greedy_{6\times6}$ with masks set $\mathbf{M}_{3\times3}$ or $\mathbf{M}_{6\times6}$ respectively. Note that $greedy_{6\times6}$ requires 71 classifier evaluations to find the worst-case masks, which is still high in computation. We address this issue with our other variant of greedy masking called *Multi-size Greedy Cutout*. As mentioned in Sec. 3.4, this variant decomposes the larger mask $100 \times 100$ mask from $\mathbf{M}_{3\times3}$ neatly into four $69 \times 69$ masks belonging to $\mathbf{M}_{6\times6}$. Henceforth, this variant approximates the worst-case masks from $\mathbf{M}_{6\times6}$ in just 26 classifier evaluations which is 2.7× faster than the former approach. We refer the masking strategy from *Multi-size Greedy Cutout* as *greedy*. The neat decomposition of masks from $\mathbf{M}_{3\times3}$ to $\mathbf{M}_{6\times6}$ allow us to use these two mask sets in our approach and therefore we adapt the same masks sets in the above masking strategies for fair comparison.

### 4.2 Results

We report our main results in Table 1 and compare the performance of our method with various prior works.

**Significant improvements in certified robust accuracy:** PatchCleanser Xiang et al. (2022) is the state-of-the-art method for certification against adversarial patches. It has been shown that it improves certification performance compared to previous methods Chiang et al. (2019); Zhang et al. (2020); Levine & Feizi (2020); Xiang et al. (2021); Metzen & Yatsura (2021) by a large margin. Table 1 shows that our proposed *Multi-size Greedy Cutout* masking strategy improves robust accuracy over the PatchCleanser Random Cutout baseline across all three classifiers (ResNet, ViT, ConvNeXt) and datasets (ImageNette, ImageNet, CIFAR-10), thus setting new state-of-the-art certified robustness results. The performance gap is noticeably increasing with increased certification pixels. Notably, certified robust accuracy on ImageNet with a ViT-B16-224 model increases from 52.5% to 57.7% against a 3% square patch applied anywhere on the image (mask set size of $3^2 = 9$). We also see large improvements for ImageNette and CIFAR-10 datasets. We perform extensive comparisons with baselines and prior work for the ImageNette, ImageNet and CIFAR-10

Table 3: **Comparing different masking strategies -** We show robust accuracy on the ResNet architecture using 3% pixel patch for ImageNette and ImageNet, and 2.4% pixel patch for CIFAR-10, CIFAR-100 and SVHN. In order from top to bottom, the sets of rows show the performance of Random Cutout with certification masks (4.1.2), Gutout - Cutout guided by Grad-CAM (4.1.3), Cutout guided by exhaustive search 4.1.4), and Greedy Cutout (4.1.5) strategies.

| Method | #passes | Mask set $\mathbf{M}_{3 \times 3}$ | | | | | Mask set $\mathbf{M}_{6 \times 6}$ | | | | |
|---|---|---|---|---|---|---|---|---|---|---|---|
| | | ImageNette | ImageNet | CIFAR-10 | CIFAR-100 | SVHN | ImageNette | ImageNet | CIFAR-10 | CIFAR-100 | SVHN |
| $\text{rand}_{3 \times 3}$ | 0 | 90.8 | 45.2 | 72.0 | 44.4 | 43.2 | 93.4 | 49.1 | 77.4 | 52.0 | 55.7 |
| $\text{rand}_{6 \times 6}$ | 0 | 89.1 | 42.5 | 65.5 | 39.1 | 33.9 | 92.7 | 49.0 | 73.8 | 48.9 | 50.4 |
| rand | 0 | 91.3 | 45.1 | 71.0 | 43.7 | 42.8 | 93.8 | 50.3 | 77.1 | 51.6 | 56.7 |
| $\text{gutout}_{3 \times 3}$ | | 92.2 | 47.0 | 74.4 | 48.4 | 40.1 | 94.0 | 50.5 | 79.1 | 55.1 | 54.4 |
| $\text{gutout}_{6 \times 6}$ | 4 | 92.0 | 44.3 | 70.3 | 42.9 | 36.3 | 94.1 | 49.3 | 76.2 | 51.7 | 49.5 |
| gutout | | 92.3 | 46.9 | 75.1 | 48.1 | 42.9 | 94.3 | 51.1 | 79.6 | 55.0 | 55.8 |
| $\text{grid}_{3 \times 3}$ | 45 | 93.7 | 51.6 | 79.7 | 54.2 | 52.8 | 95.7 | 54.9 | 84.0 | 60.1 | 62.6 |
| $\text{grid}_{6 \times 6}$ | 666 | 93.2 | - | 78.8 | 52.5 | 53.5 | 95.6 | - | 85.1 | 62.5 | 69.3 |
| $\text{greedy}_{3 \times 3}$ | 17 | 93.4 | 51.6 | 79.4 | 54.1 | 53.4 | 95.4 | 55.0 | 84.1 | 60.2 | 64.1 |
| $\text{greedy}_{6 \times 6}$ | 71 | 93.5 | 50.0 | 78.5 | 52.0 | 52.7 | 95.6 | 57.6 | 84.9 | 61.8 | 69.5 |
| **greedy (Ours)** | 26 | 93.7 | 51.2 | 79.2 | 53.9 | 54.5 | 95.6 | 56.6 | 84.3 | 61.0 | 67.5 |

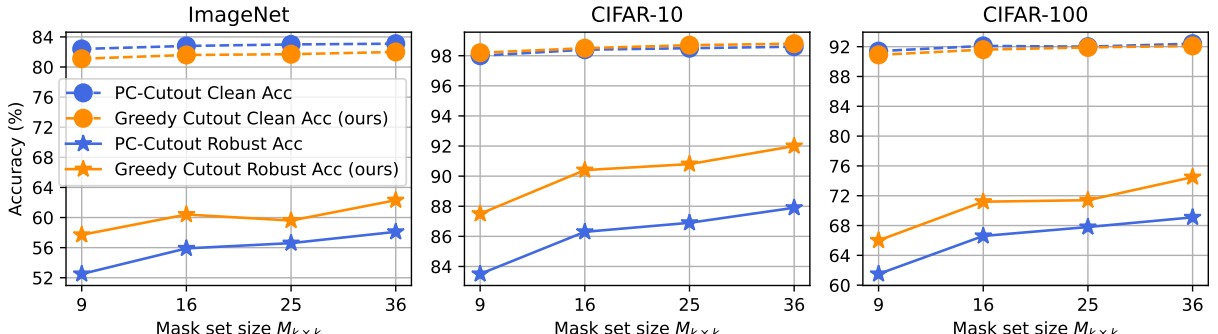

Figure 5: **Generalizability of Multi-size Greedy Cutout training to distinct certification mask set -** We train a ViT-B16-224 model using our proposed Multi-size Greedy Cutout. During training, we use masks from $\mathbf{M}_{3 \times 3}$ and $\mathbf{M}_{6 \times 6}$. During certification, we observe that this model improves certified accuracy even when mask sets $\mathbf{M}_{4 \times 4}$ and $\mathbf{M}_{5 \times 5}$ are used.

datasets. In addition, we also show that our approach improves certified robustness on additional datasets like CIFAR-100 and SVHN. In Table 2, we see that certified robust accuracy on CIFAR-100 with a ViT-B16-224 increases from 69.1% to 74.5% against a 2.4% square patch applied anywhere on the image (mask set size of $6^2 = 36$).

In Table 1, we observe that Multi-size Greedy Cutout provides a large improvement in robust accuracy in comparison to PatchCleanser, but that comes with a slight reduction in clean accuracy. For e.g., clean accuracy on ImageNet with a ViT-B16-224 decreases from 83.1% to 82.0% from the certification with mask set $\mathbf{M}_{6 \times 6}$ at 3% patch size. We believe this is an acceptable compromise for the improvement in robustness.

**Performance of different masking strategies:** Table 3 compares performance of different masking strategies discussed in Sec. 4.1 on ResNet. We observe that *Greedy Cutout* performs better than *Random Cutout with certification masks* and also *Gutout*. Moreover, it is comparable with the expensive *Grid search*. For e.g., certified robust accuracy for *Multi-size Greedy Cutout* on ImageNet is 51.2% against a 3% patch (mask set size of $3^2 = 9$). This is better than *Random Cutout* at 45.1% and *Gutout* at 46.9%. *Grid search* is slightly higher at 51.6%. Results for ViT are provided in Table A1 in the appendix.

**Tradeoff between defense performance and compute time (*Multi-size Greedy Cutout*):** We also compare the number of extra forward passes needed during training for each method. We see that the *Greedy Cutout* requires lesser compute than *Grid seach*, e.g. 71 unique forward passes compared to 666 for a $6 \times 6$ mask set, but achieves comparable robust accuracy (Table 3). Also, our *Multi-size Greedy Cutout* is an efficient strategy to reduce forward passes from 71 to only 26 but still achieve competitive certification performance with a mask set of $6 \times 6$.

**Generalizability of Multi-size Greedy Cutout training to distinct certification mask set:** We plot the robust accuracy against certification mask set size in Fig 5. We observe that robust accuracy increases when mask set size increases during certification. Our *Multi-size Greedy Cutout* maintains an improvement over all certification mask sets $\mathbf{M}_{3 \times 3}$, $\mathbf{M}_{4 \times 4}$, $\mathbf{M}_{5 \times 5}$, $\mathbf{M}_{6 \times 6}$ even though only masks from $\mathbf{M}_{3 \times 3}$ and $\mathbf{M}_{6 \times 6}$ were used during Multi-sized Greedy Cutout training.

## 5    Conclusion

PatchCleanser Xiang et al. (2022) certification requires all the double masked image combinations to be correctly classified for an image to be certified against adversarial patches. An image classifier should be invariant to pixel masking to leverage such a certification process. PatchCleanser trains models with Random Cutout augmentation. In this paper, we show that training the classifier with cutout applied at worst-case regions, i.e., regions which produce the highest classification loss, improves certification accuracy over PatchCleanser by a large margin. This indicates that our training strategy improves classifier invariance to pixel masking over prior methods. We investigate other masking strategies like Grad-CAM guided masking and exhaustive grid search to find the worst-case mask regions. To reduce the computational burden of exhaustive grid search, we introduce a two round greedy masking strategy (Greedy Cutout) to find the worst-case regions. Greedy Cutout trained models along with PatchCleanser certification sets the state-of-the-art certified robustness against adversarial patches across different models and datasets.

**Limitations:** 1. Computation: compared to random cutout, our multi-size greedy cutout increases training time with additional 26 forward passes in each training iteration. However, time for certification and defense remain same as PatchCleanser. 2. Certification: PatchCleanser requires the estimation of patch size in advance to generate the mask set and it is limited to specific patch shapes. Note that robustness guarantees are provided only to the single patch attacks but not to other form of adversarial attacks.

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

# Appendix

---

**Algorithm 1** Multi-size Greedy Cutout

---

**Input:** Image label pair $(\mathbf{x}, y)$, vanilla classifier $\mathbb{F}$, mask sets $\mathbf{M}_{3\times3}, \mathbf{M}_{6\times6}$
**Output:** Augmented data $(\mathbf{x}'_1, y), (\mathbf{x}'_2, y)$

1: **procedure** ROUND-1$(\mathbf{x}, \mathbb{F}, \mathbf{M}_{3\times3}, \mathbf{M}_{6\times6})$
2:    loss $\leftarrow -1$.
3:    **for** each $\mathbf{m} \in \mathbf{M}_{3\times3}$ **do**
4:        **if** $\ell(\mathbb{F}(\mathbf{x} \odot \mathbf{m}, y) >$ loss **then**
5:            $\mathbf{M}_1 \leftarrow \mathbf{m}$                          ▷ Mask with the largest loss
6:        **end if**
7:    **end for**
8:    $\widetilde{\mathbf{M}} \leftarrow \{4 \text{ mask from } \mathbf{M}_{6\times6} \text{ that covers } \mathbf{M}_1\}$.
9:    loss $\leftarrow -1$
10:    **for** each $\mathbf{m} \in \widetilde{\mathbf{M}}$ **do**
11:        **if** $\ell(\mathbb{F}(\mathbf{x} \odot \mathbf{m}, y) >$ loss **then**
12:            $\mathbf{M}_{11} \leftarrow \mathbf{m}$                          ▷ Mask with the largest loss
13:        **end if**
14:    **end for**
15:    **return** $\mathbf{M}_1, \mathbf{M}_{11}$
16: **end procedure**

17: **procedure** ROUND-2$(\mathbf{x}, \mathbb{F}, \mathbf{M}_{3\times3}, \mathbf{M}_{6\times6})$
18:    $\mathbf{M}_1, \mathbf{M}_{11} \leftarrow$ ROUND-1$(\mathbf{x}, \mathbb{F}, \mathbf{M}_{3\times3}, \mathbf{M}_{6\times6})$
19:    loss $\leftarrow -1$
20:    **for** $\mathbf{m} \in \mathbf{M}_{3\times3}$ **do**
21:        **if** $\ell(\mathbb{F} \odot \mathbf{M}_{11} \odot \mathbf{m}) >$ loss **then**
22:            $\mathbf{m}_3^* \leftarrow \mathbf{m}$                          ▷ Mask with the largest loss
23:        **end if**
24:    **end for**
25:    $\mathbf{M}_2 \leftarrow \mathbf{M}_1 \odot \mathbf{m}_3^*$
26:    $\widehat{\mathbf{M}} \leftarrow \{4 \text{ mask from } \mathbf{M}_{6\times6} \text{ that covers } \mathbf{m}_3^*\}$.
27:    loss $\leftarrow -1$
28:    **for** each $\mathbf{m} \in \widehat{\mathbf{M}}$ **do**
29:        **if** $\ell(\mathbb{F} \odot \mathbf{M}_{11} \odot \mathbf{m}) >$ loss **then**
30:            $\mathbf{m}_6^* \leftarrow \mathbf{m}$                          ▷ Mask with the largest loss
31:        **end if**
32:    **end for**
33:    $\mathbf{M}_{22} \leftarrow \mathbf{M}_{11} \odot \mathbf{m}_6^*$
34:    $\mathbf{x}'_1 \leftarrow \mathbf{x} \odot \mathbf{M}_2$                          ▷ Applying mask to augment data
35:    $\mathbf{x}'_2 \leftarrow \mathbf{x} \odot \mathbf{M}_{22}$                          ▷ Applying mask to augment data
36:    **return** $(\mathbf{x}'_1, y), (\mathbf{x}'_2, y)$
37: **end procedure**

---

## A   Datasets

We use five popular image classfication datasets ranging from low-resolution to high resolution images.

(1) **ImageNet:** ImageNet is an image classification dataset Deng et al. (2009) with 1000 classes. It has 1.3 million training images and 50k validation images.

(2) **ImageNette:** ImageNette Howard et al. (2020) is a 10-class subset of ImageNet with 9469 training images and 3925 validation images.

(3) **CIFAR-10:** CIFAR-10 Krizhevsky et al. (2009) is a benchmark dataset for low-resolution image classification. The CIFAR-10 dataset consists of 60k 32x32 colour images in 10 classes, with 6k images per class.

There are 50k training images and, 10k test images.

(4) **CIFAR-100:** This dataset Krizhevsky et al. (2009) is just like the CIFAR-10, except it has 100 classes containing 600 images each. There are 500 training images and 100 testing images per class. The 100 classes in the CIFAR-100 are grouped into 20 superclasses.

(5) **SVHN:** Street View House Numbers (SVHN) Netzer et al. (2011) is a digit classification benchmark dataset that contains 600,000 $32\times32$ RGB images of printed digits (from 0 to 9) cropped from pictures of house number plates.

**Models:** We use image classifiers from three different architecture families.

(1) **ResNet:** ResNets He et al. (2016) are deep neural networks which use skip connections. This approach makes it possible to train the network on thousands of layers without affecting performance. We use the ResNetV2-50x1 model from the timm Wightman et al. (2021) library.

(2) **Vision Transformers (ViT):** Convolutional Nets are designed based on inductive biases like translation invariance and a locally restricted receptive field. Unlike them, transformers are based on a self-attention mechanism that learns the relationships between elements of a sequence. We use ViT-B16-224 model.

(3) **ConvNeXt:** ConvNeXt Liu et al. (2022) is a pure convolutional model (ConvNet), inspired by the design of Vision Transformers. The design starts from a standard ResNet (e.g. ResNet50) and gradually "modernizes" the architecture to the construction of a hierarchical vision Transformer (e.g. Swin-T Liu et al. (2021)). We use the ConvNeXt_tiny_in22ft1k model from timm. It is trained on ImageNet-22k and fine-tuned on ImageNet-1k.

Table A1: Comparing certified robust accuracy of different masking strategies at two different mask set configurations $\mathbf{M}_{3\times3}$ and $\mathbf{M}_{6\times6}$ across different datasets on ViT. Certification pixels used 3% for ImageNette and ImageNet, and 2.4% for CIFAR-10, CIFAR-100 and SVHN. [1]

| | Method | #passes | Mask set $\mathbf{M}_{3\times3}$ | | | | | Mask set $\mathbf{M}_{6\times6}$ | | | | |
|---|---|---|---|---|---|---|---|---|---|---|---|---|
| | | | ImageNette | ImageNet | CIFAR-10 | CIFAR-100 | SVHN | ImageNette | ImageNet | CIFAR-10 | CIFAR-100 | SVHN |
| ViT | rand$_{3\times3}$ | 0 | 94.3 | 52.7 | 83.0 | 59.8 | 53.0 | 96.5 | 56.7 | 88.3 | 67.9 | 67.1 |
| | rand$_{6\times6}$ | 0 | 93.6 | 50.6 | 79.5 | 54.9 | 42.2 | 96.0 | 56.7 | 86.0 | 64.4 | 61.3 |
| | rand | 0 | 94.7 | 52.7 | 83.1 | 60.1 | 52.0 | 96.4 | 58.3 | 88.4 | 68.5 | 67.7 |
| | grid$_{3\times3}$ | 45 | 95.3 | 57.9 | 88.0 | 67.3 | 62.7 | 97.3 | 60.5 | 92.2 | 74.6 | 74.0 |
| | grid$_{6\times6}$ | 666 | 95.3 | - | - | - | 61.1 | 97.2 | - | - | - | 78.1 |
| | greedy$_{3\times3}$ | 17 | 95.1 | 58.3 | 87.9 | 66.6 | 62.5 | 97.2 | 61.2 | 92.2 | 74.2 | 73.8 |
| | greedy$_{6\times6}$ | 71 | 95.1 | 56.6 | 86.2 | 64.4 | 60.6 | 97.5 | 63.8 | 91.8 | 74.3 | 77.9 |
| | **greedy (Ours)** | 25 | 95.5 | 57.7 | 87.5 | 66.0 | 63.3 | 97.3 | 62.3 | 92.0 | 74.5 | 76.8 |

Table A2: Table listing the number of forward passes needed in each batch training for grid search and our *Greedy Cutout* approach.

| Method | # forward passes/batch training |
|---|---|
| grid search$_{3\times3}$ | 45 unique among $9\times9=81$ |
| grid search$_{6\times6}$ | 666 unique among $36\times36=1296$ |
| greedy$_{3\times3}$ | 9 (round 1) + 8 (round 2) = 17 |
| greedy$_{6\times6}$ | 36 (round 1) + 35 (round 2) = 71 |
| **greedy (Ours)** | 13 (round 1) + 13 (round 2) = 26 |

## B  Masks selected

Figure A1 depicts the masks selected from exhaustive cutout and our Multi-size Greedy Cutout from both $\mathbf{M}_{3\times3}$ and $\mathbf{M}_{6\times6}$ on ImageNet training samples. It can be observed in the figure that masks could potentially cover the entire object (e.g. last row) and training on such instances would limit model learning capabilities. On the contrary, we observe that this training scheme encourages for higher robust accuracy as shown in Table 3, where $grid_{3\times3}$ and our Multi-size Greedy Cutout (greedy) achieve comparable results for $\mathbf{M}_{3\times3}$ and $grid_{6\times6}$ yield higher numbers than ours on $\mathbf{M}_{6\times6}$ with significantly higher training complexity. We

---

[1]Note that training with the masking strategy $greedy_{6\times6}$ on ImageNet is computationally costly and was not possible with available GPU resources. Same for Table 3.

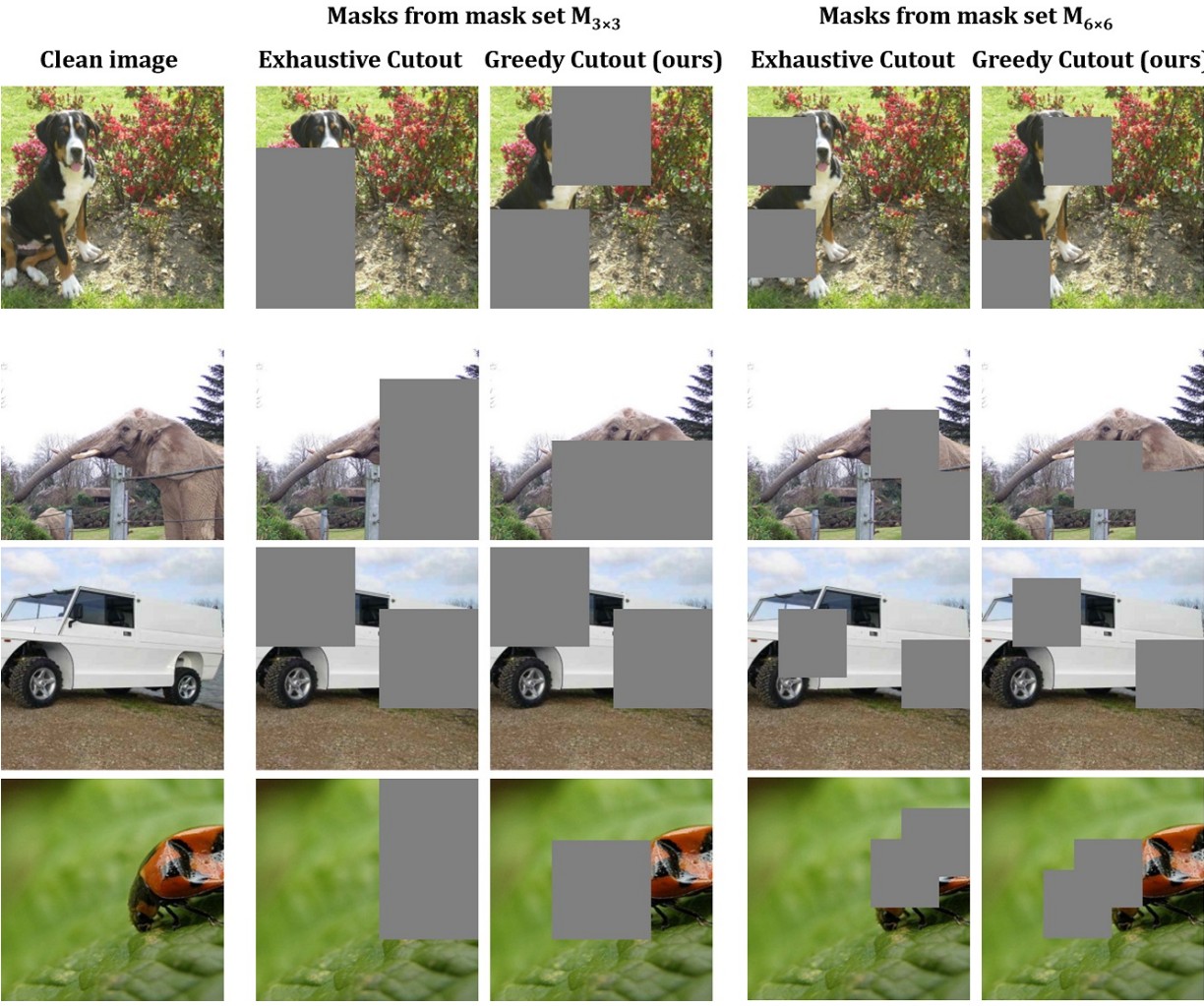

Figure A1: Masks selected via cutout guided by exhaustive search and our Multi-size Greedy Cutout from both the mask sets $\mathbf{M}_{3\times3}$ and $\mathbf{M}_{6\times6}$ on ImageNet training samples.

hypothesize that images with partially covered objects are dominating in number and hence providing a strong training signal to the model, which making the noisy training signal from fully covered objects negligible.

