# OpenReview forum: "Revisiting Image Classifier Training for Improved Certified Robust Defense against Adversarial Patches"
_TMLR — Accepted by TMLR_

### Review · Reviewer_7Aio · 2023-07-22

**Summary Of Contributions:**

This paper deals with the challenge of improving the certified robustness of image classifier against adversarial patches. The proposed method is based on the notion of worst-case masking, utilizing a greedy strategy to approximate the worst-case mask location. Experimental results show that the proposed method is superior to the previous state-of-the-art.

**Audience:**

Yes

**Broader Impact Concerns:**

I have no concerns about the ethical implications of the work.

**Claims And Evidence:**

Yes

**Requested Changes:**

In addition to the aforementioned weaknesses, I would encourage the authors to consider the following changes:

- Improve the presentation structure. I would suggest you relocate Subsection 4.1 to Section 3 and make the description of Table 3 a subsection for the ablation study of masking strategies.

- The paragraph title “Multi-size Greedy Cutout” in subsection 3.4 is repeated for describing Round 1 and Round 2. In fact, It would be better if the whole subsection 3.4 can be rewritten in a more organized way.

**Strengths And Weaknesses:**

Strengths

- The paper addresses a significant goal. Adversarial patches are more realistic threats and certified robustness is more reliable than empirical defenses.

- The motivation of the proposed method is made clear.

- The proposed method is simple and presents a degree of novelty.

- The experiments are extensive as both small and large datasets and both CNNs and ViTs are included.


Weaknesses:

- The proposed method is comparably incremental, deriving heavily from the previous work, PatchCleanser. Only one component of the algorithm exhibits significant improvement.

- Comparisons of the training time are missing. Given that the proposed method increases the computation time during training, it is important to inform readers about the extent of time investment required to improve robust accuracy. The number of passes shown in Table 3 does not sufficiently convey this information.

- The structure could use some refinement. For example, Subsection 4.1 is in the Experiments section, yet it does not contain any experimental results.

---

### Review · Reviewer_G1gq · 2023-07-28

**Summary Of Contributions:**

This paper proposes a simple Greedy Cutout strategy to improve doubled-masks of PatchCleanser. The main idea is to to choose worst-case masked images with high classification loss for training. The worst-case masks are chosen from the mask set that are used during certification.

**Audience:**

Yes

**Broader Impact Concerns:**

The paper proposes a technique to certify the patch attack. I do not have any concern of the ethical implications.

**Claims And Evidence:**

No

**Requested Changes:**

1) The authors mentions that " PatchCleanser uses Random Cutout augmentation DeVries & Taylor (2017), i.e., applying two masks of size 128×128 at random locations to 224 × 224 training image", which seems different from what were mentioned to in Alg. 2 and Sec. 4 in that paper.

2) Why are the results of PatchCleanser presented in this paper different from those in the original paper (cf. Table 2 in the original paper)?

3) The authors should show the running times of the proposed approach and the baselines.

4) As shown in Alg. 2 in the PatchCleanser, for $M_{3 \times 3}$, the number of pairs $m_0, m_1$ to go through is $36$ which is not a high number. Therefore, it is unclear to me the motivation of this work.

**Strengths And Weaknesses:**

Strengths:
- The idea is simple and intuitive.
- The Greedy Cutout strategy outperforms other Cutout strategies

Weakness:
- This work mainly bases on PatchCleanser, hence its novelty is limited.
- The results of PatchCleanser presented in this paper are different from those in the original paper (cf. Table 2 in the original paper).
- There is no any reporting results about the running times.

---

### Review · Reviewer_BGXY · 2023-08-02

**Summary Of Contributions:**

This paper proposes a masking strategy - Greedy Cutout, for improving certified robustness against patch attacks. Greedy Cutout is designed to be used jointly with PatchCleanser. The high-level intuition is to search for the patch incurring the largest loss and then apply it. Several optimizations such as state-wise greedy search and stage-wise patch decomposition are introduced. Experiments on ImageNette, ImageNet, and CIFAR-10 with multiple architectures show the effectiveness.

**Audience:**

Yes

**Broader Impact Concerns:**

The authors may use a new section to emphasize the limitations of patch certification in PatchCleanser, e.g., no guarantee for patch attacks larger than the given threshold or different from the supported shape, and no guarantee for other forms of attacks. So practitioners would not over-interpret the certification statement.

**Claims And Evidence:**

Yes

**Requested Changes:**

1. The manuscript's clarity could be improved by explaining new concepts in detail:
- In Figure 1(b), authors show a full occlusion can make the classifier hard to predict, while partial occlusion is more desired. This concept is restated in Section 4.1. However, how does Greedy Cutout succeed in providing partial occlusion? It seems that Greedy Cutout searches for the patch location of the highest loss, which can still occlude the object completely.
- In Section 4.1, what does "pixel masking" mean? Does it mean applying the mask at the pixel-level granularity?
- In Section 4.1.4, what does "long edges" mean? Why does exhaustive search induce "long edges"?

2. Could the authors discuss the generalizability of the approach for other sets of patch masks beyond $M_{3\times 3}$ and $M_{6\times 6}$? The illustration appears to be very specific. It would be great if some high-level guidance is provided in this sense.

3. Though Greedy Cutout improves the efficiency greatly compared to exhaustive search, in experimental evaluation, there is no training time control. A comparison with PatchCleanser under the same training time budget would be great, e.g., by setting different training epochs for different methods.

**Strengths And Weaknesses:**

Strengths:
- The motivation and method design is reasonable, simple yet effective. The key insight behind Greedy Cutout is akin to adversarial training in my opinion, which has been widely received to improve the robustness effectively. In the patch robustness case, it leads to the search for the patch incurring the largest loss. The optimizations are novel and smart.
- Strong empirical performance. Across a wide range of datasets and architectures, the proposed approach improves the certified accuracy significantly with little loss of normal accuracy.

Weaknesses:
- The presentation could be further improved. Though there are few grammatical or writing issues, some concepts and logic appears unclear to me. See details in "Requested Changes".
- The proposed method could be limited in terms of needing a specific patch shape and specific certification method.
- The experimental evaluation lacks the comparison under the same running time with published baselines.

---

### Review · Reviewer_QQsV · 2023-08-03

**Summary Of Contributions:**

This paper presents an enhanced method building upon PatchCleanser. The core concept introduces the notion that the classifier, when trained using the worst case masked images, yields optimal classification outcomes. The paper outlines a two-step greedy approach for identifying these complex masks. Experimental results demonstrate notable improvement across multiple datasets and models.

**Audience:**

Yes

**Broader Impact Concerns:**

I have no concerns about the ethical implications of the work.

**Claims And Evidence:**

Yes

**Requested Changes:**

1. To enhance the support for the greedy cutout method, the author could provide a direct comparison between masks selected using the greedy cutout approach and those selected via exhaustive cutout.
2. The paper's value could be significantly demonstrated by including a table that outlines computation times for various methods.
3. To bolster the intuitive concept behind exhaustive search cutout, the author is encouraged to delve deeper into its rationale through additional discussion.
4. Language Modifications:
   a) Page 1, second paragraph of the Introduction, line 5: Remove "for" from "Hence. the certified robust accuracy for is the".
   b) Page 2, line 1: Change "proposes two round pixel masking" to "proposes two-round pixel masking" or "proposes two rounds of pixel masking".
   c) Page 5, second paragraph, line 1: Remove "of" from "In the first round of". Please also review the paper for other potential language improvments.



**Strengths And Weaknesses:**

Strengths:
1. The proposed intuitive greedy-cut method closely approximates classifier accuracy achieved through exhaustive cutout search, highlighting its effectiveness.
2. A notable advantage of the greedy-cut approach is its reduction in training time.
3. Remarkable enhancements in accuracy across various datasets are demonstrated through the application of the greedy-cut method.

Weaknesses:
1. Doubts emerge concerning the validity of employing the exhaustive search method as an upper limit, as it tends to result in complete occlusion of the target. This renders the classification task inherently impractical, potentially leading to limited learning for the model.
2. The significant contribution of this paper, which pertains to improved time efficiency, lacks comprehensive elaboration.
3. Certain language issues are present within the paper that require attention and refinement.

---

> ### Author Response · Authors · 2023-08-16
> **Authors rebuttal**
>
> >Direct comparison between masks selected using greedy cutout and those selected via exhaustive cutout.
>
> Thank you for the suggestion. We have included qualitative comparison of the selected masks of our greedy cutout and exhaustive cutout in Figure A1 (Appendix B) in the supplementary material. Quantitative comparison between these strategies can already be found in Table 3 (exhaustive cutout $grid_{3\times3}$ and $grid_{6\times6}$ are compared against our greedy approach).
>
> >Doubts on employing the exhaustive search method as an upper limit, as it tends to complete occlusion of the target.
>
> We agree with the reviewer that exhaustive search could potentially yield complete occlusion of the object (e.g. last row of Figure A1 in supplementary material). This is particularly true for the larger masks from $M_{3\times3}$ and weakly holds for $M_{6\times6}$ with its smaller size of masks. We also notice in Table 3 that our greedy method reach comparable performance with $grid_{3\times3}$, while $grid_{6\times6}$ on mask set $M_{6\times6}$ consistently outperform our approach across datasets.  These results suggest that exhaustive search with smaller mask set would serve as better approximates for the upper bound. Therefore, reliability of the upper bound from the exhaustive cutout improves as the masks get smaller in size. However, computing the upper bound with mask set much smaller than $M_{6\times6}$ is computationally expensive. Hence, we favor exhaustive cutout with $M_{6\times6}$ as a reasonable choice for approximating the upper bound.
>
> >Intuitive concept behind exhaustive search cutout
>
> Correct prediction of all possible two-masked images from mask set $M_{k\times k}$ is essential for PatchCleanser certification. Here, invariance to pixel masking is of vital importance for achieving better certification results. Exhaustive search is considered to mimic the "certification time processing of the two-masked images" but during training. Through the search, we aim to find the hard cases for the model and train the model on such cases to improve its invariance to pixel masking.
>
> >Table that outlines computation times
>
> Please refer to our global response above for the computation time table.
>
> >Language Modifications
>
> Thank you for pointing out these modifications. We have incorporated your suggestions in our revised manuscript.

---

### Decision · Action_Editors · 2023-09-05

**Recommendation:** Accept as is

**Comment:**

This paper proposed a new and effective method for certified training against adversarial patch attacks. The novelty lies in the proposed two-round greedy masking strategy (Greedy Cutout) for approximating worst-case mask location. Experimental results show improved certified robust accuracy with a notable increase when compared to state-of-the-art. The authors' rebuttal has successfully addressed the reviewers' questions. All reviewers recommend to accept this submission, and I concur.

**Audience:**

The topic is certified robustness, which is of broad interest to TMLR audience

**Claims And Evidence:**

The claims are well supported by the experiments